# Effects of Practicing Closed- vs. Open-Skill Exercises on Executive Functions in Individuals with Attention Deficit Hyperactivity Disorder (ADHD)—A Meta-Analysis and Systematic Review

**DOI:** 10.3390/bs14060499

**Published:** 2024-06-14

**Authors:** Chunyue Qiu, Qun Zhai, Shuangru Chen

**Affiliations:** Faculty of Health Sciences and Sports, Macao Polytechnic University, Macao SAR, China; p2214177@mpu.edu.mo (C.Q.); p2316799@mpu.edu.mo (S.C.)

**Keywords:** attention deficit disorder with hyperactivity, inhibitory control, cognitive flexibility, working memory, meta-analysis

## Abstract

(1) Background: Previous studies have identified discrepancies in improvements in executive functioning in typically developing children when comparing closed- and open-skill exercise interventions. However, there is limited research on executive functioning in attention deficit hyperactivity disorder (ADHD). This study aims to conduct a systematic review and meta-analysis to explore the impact of closed- and open-skill exercises on ADHD populations. (2) Methods: The PRISMA guidelines for systematic reviews were followed to search seven databases to evaluate and analyze studies published from 2013 to 2023. Prospero: CRD42023460452. (3) Results: A meta-analysis of 578 subjects with ADHD in 11 RCTs (Randomized control trial) and 3 NRS (Non-randomized studies) revealed that closed-skill exercise significantly improved executive function subdomains, including inhibitory control (standardized mean differences (SMD) = −1.00), cognitive flexibility (SMD = −1.33), and working memory (SMD = −0.85). Furthermore, open-skill exercise was found to have a positive effect on inhibitory control (SMD = −1.98) and cognitive flexibility (SMD = −0.97) in ADHD patients. Both types of exercise interventions demonstrated an improvement in executive function compared to controls, with open-skill exercises exhibiting superior effects (Q_b_ = 6.26). (4) Conclusions: The review recommends a 12-week intervention cycle with exercise at least twice a week of moderate or higher intensity as suitable for ADHD individuals. This review also encourages individuals with ADHD to engage in exercises involving multiple motor skill types.

## 1. Introduction

Attention deficit hyperactivity disorder (ADHD) is one of the most common childhood psycho-behavioral disorders, characterized by pervasive and impairing symptoms of inattention, hyperactivity, and impulsivity [1]. Children with ADHD have persistent functional deficits in cognitive response inhibition and vigilance, working memory, and planning. The severity of early ADHD affects clinical and academic development in adulthood, and ADHD is associated with the severity of suicidal behavior in adulthood [2,3]. The prevalence varies considerably around the world, but it is generally accepted that the global prevalence is 5.29%, with 60–80% of children with the disorder experiencing symptoms that persist into adulthood [4,5]. The prevalence in Chinese adolescents is 6.3%, and the true prevalence may be even higher [6]. There are pharmacological and non-pharmacological treatments for ADHD, respectively. Although pharmacological treatments are therapeutic, there are varying degrees of side effects associated with each medication, which include serious side effects such as the potential for growth inhibition, the exacerbation of psychiatric disorders, abnormal movements, and obsessive compulsive behaviors [7]. Non-pharmacological treatments are more time-consuming but do not result in adverse effects. As a result, non-pharmacological treatments have been given more attention and have been promoted globally [8].

ADHD primarily affects prefrontal lobe functions of the brain, including executive function (EF), attention, memory, cognitive control, time management, and emotion regulation [9]. Deficits in executive functioning are the most common of ADHD symptoms, accounting for more than two-thirds of patients’ symptoms and having one of the most significant impacts on patients’ daily lives [10]. The three widely accepted subfunctions of executive function are inhibition, working memory, and cognitive flexibility [11]. Children with ADHD may encounter difficulties in processing incoming information, identifying critical information, controlling interfering thoughts, remembering and associating information, and maintaining focus on tasks during class due to deficits in EF [12]. Although EF deficits are not the only cognitive and behavioral problems in children with ADHD, they have a significant impact on the symptoms of the disorder, and improving EF can be effective in improving ADHD symptoms [13]. Many studies have confirmed that exercise can improve various aspects of cognition and performance [14,15,16]. One study showed that aerobic exercise is associated with better performance control tasks and better performance outcomes [17]. Exercise interventions are also increasingly emphasized in the treatment of ADHD, and although evidence is limited, current research generally supports the role of acute chronic physical activity (PA) in reducing ADHD [18]. Some findings show that a single session of moderate-intensity aerobic exercise positively affects neurocognitive function and inhibitory control in healthy and ADHD children [19]. It has also been demonstrated that higher levels of physical activity are associated with better EF performance in children with ADHD [18]. The findings of Liang et al. suggest that exercise interventions improve overall executive functioning in children with ADHD and that exercise interventions have a moderate to large positive impact on inhibitory control and cognitive flexibility [20].

Different types of movement have varying effects on executive function. Open-skill exercises (OSEs) and closed-skill exercises (CSEs) are two distinct types of movements, differing in terms of the movement environment and requirements. OSE involves movements performed in an unrestricted external environment, often necessitating adaptation to uncertain environmental factors and requiring dynamic balance, real-time reactions, and decision-making abilities. On the other hand, CSE is performed in a restricted internal environment, typically within fixed settings and rules, with relatively stable movement requirements [21,22,23]. Research suggests that the cognitive demands of OSE may help explain the beneficial effects of exercise on inhibitory control, indicating the crucial role of environmental complexity in exercise training for children’s cognitive and motor development [24]. As the impact of physical activity on the executive function of children, particularly those with cognitive impairment, has gained increasing scholarly attention and the results of related research have begun to emerge, some scholars have conducted a review and meta-analysis of these findings. There are meta-analyses on typically developing children showing that OSE is superior to CSE in improving all aspects of EF subfunctions, with the intervention effect of OSE being significantly greater than CSE, especially in the domain of working memory [25,26]. Another study concluded that CSE is more effective on inhibitory control and OSE is more effective on working memory as well as on cognitive flexibility in a low-frequency intervention [27]. However, due to the typically weaker motor skills in ADHD patients, the findings from studies on typically developing children cannot be directly extrapolated to ADHD patients. Particularly, individualized and specialized considerations are needed in motor interventions for ADHD patients. Huang et al.'s findings indicate that prolonged exercise interventions have modest to moderately beneficial impacts on executive functioning in children with ADHD. They also highlight a significant improvement in core ADHD symptoms through closed-ended skill practice [28]. While these studies affirm the positive effects of exercise intervention on children’s executive function, they do not delve deeply into the correlation between various exercise modalities and specific subfunctions of executive function. Furthermore, the ideal dose, duration, and type of exercise for individuals with ADHD lack strong empirical support.

Given that the effects of closed- and open-skill exercises on executive function subfunctions in the ADHD populations are still unclear, this study aims to conduct a systematic review and meta-analysis to explore the impact on ADHD populations. The goal is to provide scientific evidence for more effective treatment of ADHD through exercise prescription.

## 2. Methods

This study complies with the PRISMA (Preferred Reporting Items for Systematic Reviews and Meta-Analyses) statement [29] and the systematic evaluation program is registered with international prospective register of systematic reviews PROSPERO (CRD42023460452).

### 2.1. Retrieval Strategy

In September 2023, the terms we identified after discussion and keyword aggregation from previous reviews, including combinations of subject terms and free words against the database’s official list of subject terms, ultimately constituted the search strategy and were used in PubMed, Embase, Web of Science, American psychological Association PsycINFO (via EBSCO Information Services), Cochrane Library, China National Knowledge Infrastructure (CNKI), and Wan Fang, based on the method highlighted by Field et al. in the databases after correcting the preparation for the search (Field and Gillett, 2010). We determined the subject words and free words for the search, e.g., (1) population (study population): ‘Attention Deficit Disorder with Hyperactivity’ OR ’ADHD’; (2) intervention (intervention): ‘Exercise’ OR ‘Physical Activity’; (3) outcome (outcome indicator): ‘Executive Function’ OR ‘Executive Functions’. The results were searched using “AND”, “OR”, and “NOT”. The search results were restricted to English vs. Chinese and studies from 2013 to 2023. Independent back-to-back searches were conducted by two authors (Appendix A).

### 2.2. Inclusion and Exclusion Criteria

Inclusion and exclusion criteria were developed according to the PICOS (patient, intervention, comparison, outcomes, and study design) principles to identify eligible articles for this systematic evaluation, as well as meta-analysis [30].

#### 2.2.1. Inclusion Criteria

Patients or population: people with ADHD who have been diagnosed by a doctor or who meet the criteria of the International Classification of Diseases (ICD), the Chinese Classification of Mental Disorders (CCMD), and the National Diagnostic and Manual of Mental Disorders (DSM).Type of intervention: acute or prolonged closed- or open-skill exercise interventions of moderate or higher intensity with a defined exercise program.Type of contrast: ADHD patients who are controlled by physical activity, with baseline values consistent with the experimental group.Type of outcomes: reporting outcome measures for neurocognitive tasks using executive function (EF) (i.e., Stroop test, N-Back task, and odd–even size test, etc.)Type of studies: RCT (Randomized control trial) or NRS (Non-randomized studies).

#### 2.2.2. Exclusion Criteria

Patient has co-morbidities or an abnormal IQ, i.e., IQ below 80.Studies applying other combined interventions (e.g., exercise plus nutrition program, use of electronics for exercise).Based on observational studies (i.e., cross-sectional studies, qualitative studies, letters, news reports, guidelines, conference abstracts, dissertations, etc.).Full text was not available, or data could be extracted for studies without available data, and studies lacked pre-test or post-test data.

### 2.3. Data Extraction and Classification

#### 2.3.1. Data Extraction

Two researchers (Qiu and Chen) independently screened and extracted data using EndNote 21 (Bld 19023) to ensure accurate article screening. They first read the title, abstract, and keywords, then obtained and evaluated the full text in detail, and finally extracted the data according to a pre-determined standardized table. The two researchers completed the above work independently and then imported it into the document for review. Any differences of opinion were resolved through discussion after reading the full text. If necessary, a third senior researcher (Zhai) was consulted for decision making. The final decision was made based on the input from all three researchers.

#### 2.3.2. Classification

The inclusion studies were categorized into closed and open motor skills based on the characteristics of the closed and open motor skills (Appendix B).

### 2.4. Quality Assessment

The two researchers independently assessed the quality of articles included in randomized controlled trials using the Cochrane Risk of Bias tool (RoB 2) by the Cochrane Collaboration meta-analysis guidelines. The risk of bias was assessed, with assessments focusing on randomization, allocation concealment, blinding, masked presence, and reporting bias [31], while the Risk of Bias in Non- randomized Studies - of Interventions (ROBINS-I) tool was employed to evaluate the risk of bias in non-randomized trials. The risk of bias judgments in ROBINS-I included pre-, at-, and post-intervention domains [32]. If unanimity was still not possible, a third senior researcher (Zhai) would make the decision.

### 2.5. Meta-Analysis

Stata MP 15.1 was used to analyze the data. Statistical analyses involved calculating effect sizes, including mean (mean), standard deviation (sd), and sample size (n). Hedges g was used to calculate effect sizes as it corrects for small sample size bias [33]. Effect sizes were categorized as small (<0.2), medium (0.5), and large (>0.8) [34]. Continuous variables were expressed as standardized mean differences (SMD), with 95% confidence intervals (CI). When there were multiple data in the included studies, the study data were extracted or combined according to the recommendations in the Cochrane Guidelines. When the results are not in the same direction, we x − 1 the result to unify the directions [31]. Subsequently, the heterogeneity of each study was tested, where I-squared (I^2^) < 50% indicated low heterogeneity, using a fixed-effects model, and where I^2^ > 50% indicated high heterogeneity, using a random-effects model. The study used the intuitive funnel plot, as well as the quantitative Egger’s test and Begg’s test to detect publication bias. Sensitivity analyses were conducted using the trim and fill method to explore the stability of the results when *p* < 0.05 indicated the presence of publication bias.

### 2.6. Moderator Analysis

Comprehensive Meta-Analysis Version 3.0 was used for moderated effects analysis. Moderator analysis was performed to determine whether the effect of the intervention on different executive function subfunction outcomes was due to the type of motor skill intervened (CSE or OSE), the frequency of the intervention (≤2 sessions/week or >2 sessions/week), the duration of the intervention period (<12 weeks or ≥12 weeks), and the age (<18 or >18 years). Due to the small number of included studies in which subjects were adults, only one moderator analysis of the intervention on executive function outcomes was performed, without subdividing their subfunctions.

## 3. Results

### 3.1. Literature Selection Results

Overall, there were 14 articles included in the review (Figure 1).

### 3.2. Eligible Research Features

Fourteen studies were included in the analysis, consisting of 11 RCTs [35,36,37,38,39,40,41,42,43,44,45] and three NRS [46,47,48]. Of these, nine employed closed motor skills interventions, while the other five implemented open motor skills interventions. The study sample included 578 patients diagnosed with ADHD, with 286 patients assigned to the intervention group and 292 to the control group. Most patients (70%) were male, with a total of 410 male patients. Six out of the fourteen articles included reported the number of lost or withdrawn subjects, totaling 35 (30 for the closed exercise intervention and 5 for the open exercise intervention), with a loss rate of 6% [37,40,41,43,46,47]. The exercise intensity was moderate or higher, with interventions occurring 1–5 times per week. The duration of single interventions ranged from 20 to 70 min, and the total duration of interventions ranged from 30 min to 7800 min (Table 1).

### 3.3. Methodological Quality Evaluation

In the 11 RCTs, the risk of bias was primarily associated with the randomization process. This was primarily because some of the studies lacked sufficient detail regarding the method of randomization. Additionally, in the design of exercise interventions, blinding the participants was not feasible. Figure 2 shows two studies that have a high risk of bias [35,43]. The overall risk of bias was moderate in all three NRS. Among them, the risk of bias exists mainly in the three areas of bias due to confounding and missing data, and the measurement of outcomes. (Table 2).

### 3.4. Meta-Analysis Results

#### 3.4.1. Closed Skill Types

A total of nine studies with a closed-skill exercise intervention were included in the meta-analysis. Data from 402 ADHD individuals were included. Figure 3 shows the results, respectively: the closed-skill exercises intervention had large effects on executive functioning (g = −1.04) as well as its sub-functions, inhibitory functioning (g = −1.00), cognitive dexterity (g = −1.33), and working memory (g = −0.85) in the ADHD population.

#### 3.4.2. Open Skill Types

A total of five studies with an open-skill exercise intervention were included in the meta-analysis. Data from 176 ADHD individuals were included. Figure 4 shows the results, respectively: the open-skill exercises intervention had a large effect on executive functioning (g = −1.37) and its subfunctions, inhibitory functioning (g = −1.99) and cognitive dexterity (g = −0.97), whereas the results for the working memory subfunction were not significant (g = −0.48).

### 3.5. Moderation Effect Analysis

The results showed that an OSE intervention was significantly better than a CSE intervention for improving inhibitory function in ADHD patients (Q_b_ = 6.263, *p* < 0.05). An intervention frequency of more than two sessions per week was significantly better for working memory improvement than a low intervention frequency (Q_b_ = 7.082, *p* < 0.05), and improvement in working memory subfunctions was found to be non-significant for exercise interventions at low intervention frequencies (SMD = −0.243; 95% CI [−0.673, 0.186]; *p* = 0.267). Intervention cycles longer than 12 weeks had a more notable impact on improving inhibitory function (Q_b_ = 7.587, *p* < 0.05). Although the exercise intervention was more effective in improving executive functioning in adolescents than in adults (Q_b_ = 7.031, *p* < 0.05), it was also found to be significant in the adult subgroup (SMD = −0.613; 95% CI [−1.676, −0.934]; *p* < 0.05) (Table 3).

### 3.6. Sensitivity Analysis

The studies on inclusion were classified and analyzed using the cut-and-patch method (metatrim). The results showed that none of the 95% CIs deviated from the original intervals, indicating stability in the analysis (Appendix C).

### 3.7. Tests for Publication Bias

The two groups of studies were analyzed separately for publication bias. The funnel plot of CSE was relatively symmetrical, and the results of Egger’s test (*p* = 0.402; t = −0.86) and Begg’s test (*p* = 0.893) indicated a low risk of publication bias. The funnel plot of OSE suggested some publication bias, and the results of Egger’s test (*p* = 0.059; t = −2.33) and Begg’s test (*p* = 0.386) showed a low risk of publication bias (Appendix D).

## 4. Discussion

To our knowledge, this study is the first to examine the impact of closed and open motor skills on different subfunctions of EF in patients with ADHD, including adult participants. ADHD is often thought of as a condition affecting only children and adolescents. However, the American Psychiatric Association has made it clear that adults can also experience symptoms of ADHD [1]. The meta-analysis results support the effectiveness of both closed and open exercise in improving EF in patients with ADHD. OSE was significantly more effective than CSE in improving inhibitory function in patients with ADHD (Q_b_ = 6.263, *p* < 0.05), with an OSE intervention effect value, SMD, −2.04. However, the effect on working memory showed improvement but was not significant. Exercise intervention improved executive function in adult ADHD patients, with a moderate effect value (SMD = −0.61). We found that an intervention cycle of more than 12 weeks and an intervention frequency of more than two times per week of moderate or higher intensity exercise may be an appropriate dose of exercise for the ADHD population and that people with ADHD are encouraged to participate in exercise interventions for multiple motor skill types. Our study further strengthens the evidence base for exercise as a treatment option for individuals with ADHD. 

The results of the current study are consistent with the previous narratives of Liang et al. and Huang et al. Exercise interventions have the effect of improving executive functioning in the ADHD population, and the present study broadens the population for which exercise interventions are appropriate as well as subdividing the beneficial effects of exercise skill interventions on adolescent EF subfunctions [20,28]. Heilmann et al. and Feng et al. concluded that OSE intervention was more favorable for enhancing executive function, particularly inhibitory functioning, in typically developing children. This finding is consistent with our own, which showed that moderate-intensity OSE intervention was significantly better for inhibitory functioning than CSE intervention in people with ADHD. Furthermore, it is worth noting that the study conducted by Feng et al. suggests that OSE is more effective than CSE in improving working memory in typically developing children. However, this finding is inconsistent with the results of the present study, which did not find a significant effect of OSE on working memory in the ADHD population. This discrepancy may be due to differences in the study populations and the limited number of studies available in this area [26,49]. The study by Verburgh et al. demonstrated the positive impact of exercise intervention on executive function in adult ADHD patients. Our study supports these findings, although we included participants up to the age of 50, compared to Verburgh et al.’s inclusion of participants up to 35 years old. Notably, we found a moderate effect in the 18+ age subgroup, expanding the age range for which exercise intervention may be beneficial for ADHD patients [50].

The results of this meta-analysis indicate that both CSE and OSE significantly improved executive functioning in ADHD patients, with large effect sizes. This may be due to the inclusion of studies with moderate to above-moderate exercise intensities, as guidelines for physical activity and cognition suggest a stronger association between moderate-to-vigorous-intensity PA and cognitive improvement [51]. The study found that CSE had above-moderate effect sizes for all three subfunctions of EF in ADHD patients. This may be because CSE intervention subjects were in a stable and predictable environment, allowing them to pace themselves and experience a lower cognitive load. As a result, ADHD patients were better able to acquire motor skills and maintain moderate intensity, resulting in increased cardiorespiratory fitness. Cardiorespiratory fitness is positively correlated with neuroelectric indices of attention, working memory, and reaction speed [52]. CSE may result in increased brain perfusion, particularly in the prefrontal and parietal regions [53]. Research has shown that aerobic exercise is linked to an increase in gray matter volume and the hippocampus. Additionally, increased hippocampal volume is associated with higher serum levels of BDNF (brain-derived neurotrophic factor) [54,55]. The OSE intervention was found to be more effective than CSE in increasing brain BDNF levels in the general young population [56]. However, it is generally observed that motor ability is lower in individuals with ADHD. Therefore, the elevated EF may be due to strenuous exercise. This also explains the possible reasons for the steady elevation of CSE on the various subfunctions of EF. The OSE intervention had no significant impact on working memory subfunctions and may be less effective in improving subjects’ low arousal levels due to its difficulty and intermittent nature [57]. However, the large effect size on inhibitory function may be due to the fact that, in OSE, subjects need to constantly respond to environmental changes, which increases the need to perceive and process environmental stimuli. Findings from Takahashi et al. and Visser et al. suggest that, in OSE, compared to CSE, interventions induce increased activity and neurological efficiency in prefrontal and frontal, Centro–central brain regions [58,59]. There is a correlation between brain activation and cognitive performance [60]. The better enhancement of inhibitory function in OSE compared to CSE may be due to the constant stimulation of different receptors in the subjects during the intervention, which places a greater demand on cognition and enhances inhibitory function through consistent practice. In our study, we found that a moderate-intensity CSE intervention showed a moderate effect on EF in 18- to 50-year-old ADHD patients. The potential reason for this finding may be related to certain structural cortical, cerebellar, and subcortical abnormalities in the brains of ADHD patients and the link between structural brain integrity and ADHD symptoms in both adult and immature patients, which allows for the continued ability of adult ADHD patients to enhance EF through exercise [60]. Our meta-analysis further supports a strong relationship between exercise and structural brain improvement in ADHD patients. Our finding in the moderated effects analysis that longer intervention cycles may have a better EF-enhancing effect may be related to more consistent and diverse changes in white and gray matter over longer durations [61], providing rationale for the choice of exercise intervention treatment for ADHD patients.

The present study has some limitations. (1) Only exercise intervention studies with moderate or higher exercise intensities were included in this study, although we chose the exercise intensities that were most favorable for cognitive improvement. However, we should be cautious about extrapolating our results directly to all intensities of exercise interventions. (2) The risk of bias for the included studies in this paper was high. This may be because the authors of the included studies did not comprehensively report on the randomization methodology and the blinding strategies. OSE and CSE interventions show a different focus on improving outcomes for people with ADHD, but this may need more research data to be better justified.

## 5. Conclusions

The systematic review and meta-analysis suggest that an intervention cycle lasting more than 12 weeks and an intervention frequency of more than twice per week of moderate- or higher-intensity exercise may be an appropriate exercise dose for the ADHD population. The study found that closed-skill exercise improved executive function performance in adult ADHD patients. Therefore, it is recommended that people with ADHD participate in exercise interventions for multiple motor skill types.

## Figures and Tables

**Figure 1 behavsci-14-00499-f001:**
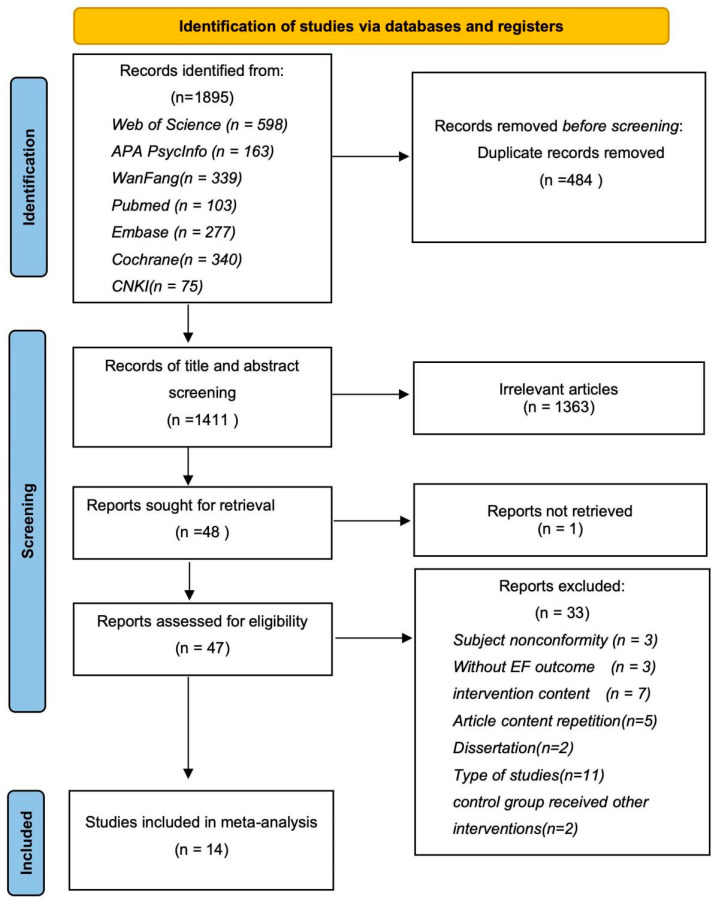
The PRISMA flow chart.

**Figure 2 behavsci-14-00499-f002:**
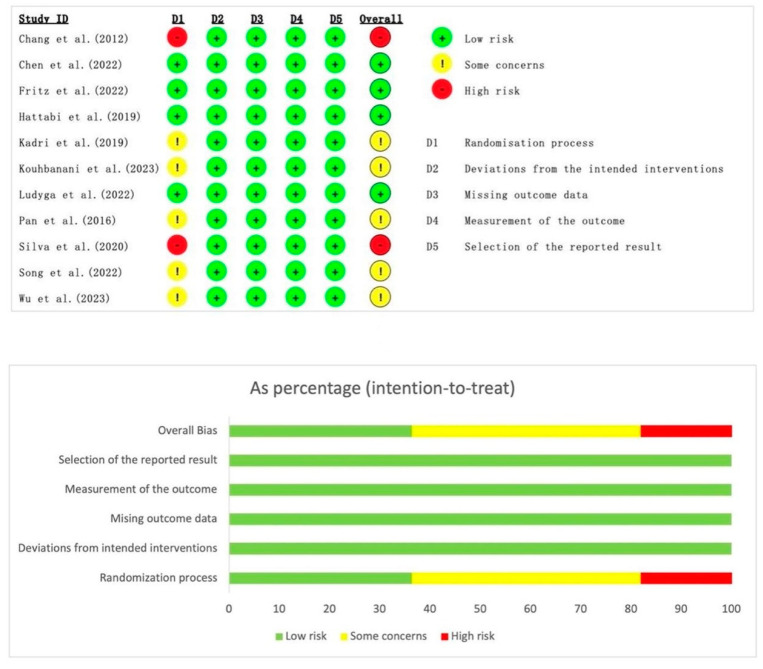
RoB2 methodological quality evaluation results [35,36,37,38,39,40,41,42,43,44,45].

**Figure 3 behavsci-14-00499-f003:**
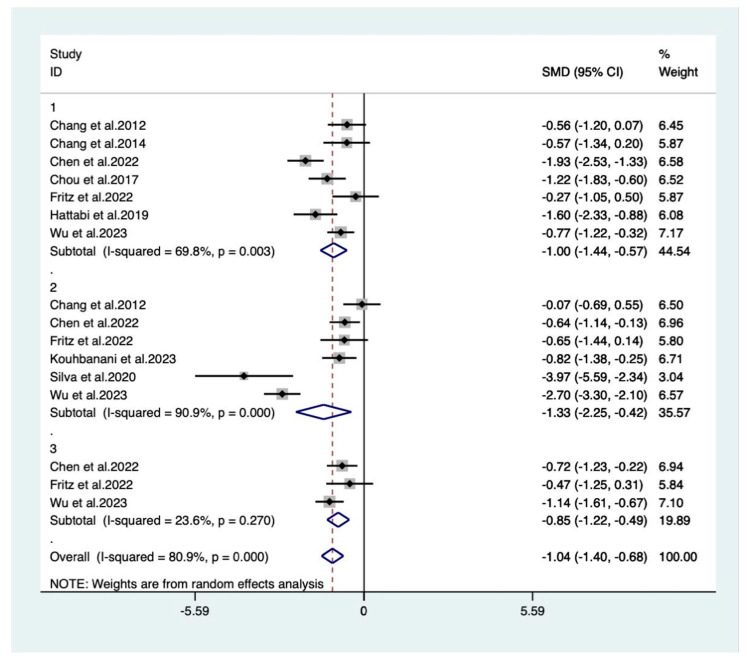
A meta-analysis of the results of closed-skill exercises on executive function in ADHD patients. Panel 1: inhibitory control; panel 2: cognitive flexibility; panel 3: working memory. Rhombus: pooled effect size; *X*-axis: solid line is the invalid line (intersection with which is a statistically non-significant difference); dashed line: value of the overall pooled effect size [35,36,37,38,40,43,45,46,47].

**Figure 4 behavsci-14-00499-f004:**
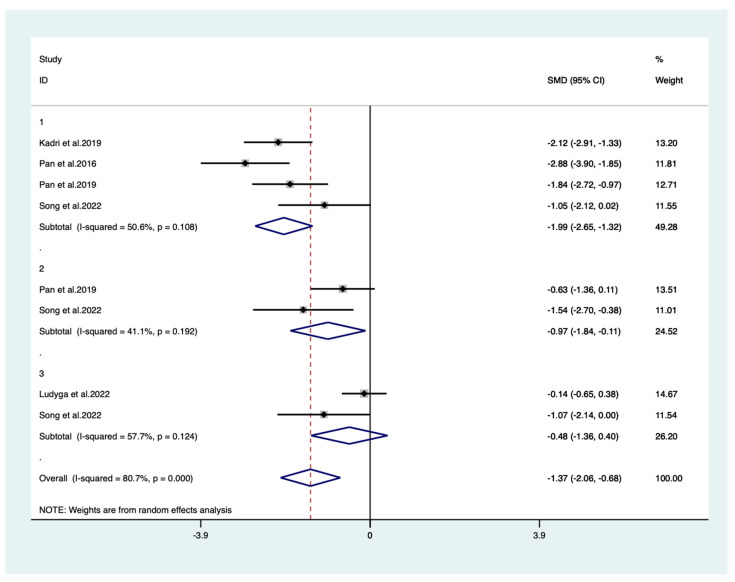
A meta-analysis of the results of open-skill exercises on executive function in ADHD patients. Panel 1: inhibitory control; panel 2: cognitive flexibility; panel 3: working memory. Rhombus: pooled effect size; *X*-axis: solid line is the invalid line (intersection with which is a statistically non-significant difference); dashed line: value of the overall pooled effect size [39,41,42,44,48].

**Table 1 behavsci-14-00499-t001:** Characteristics of included studies (closed skill).

Author (Year)	Country /District	Study Design	Participant Characteristics					Intervention Component			Outcome
			Age Range; Sex-M (%); Diagnostic Methods	Sample Size (IG/CG)	Age (Control)	ADHD Subtypes (Control)	Medication Situation (Control)	Content (Control)	Intensity	Frequency	Outcome Measures
Chang (2012) [35]	Taiwan	RCT	8–15; M-92; DSM-IV	40 (20/20)	10.45 ± 0.95 (10.42 ± 0.87)	I:10, HI:2, C:8 (I:4, HI:3, C:13)	10 (10)	Treadmill (watching videos)	Moderate (50–70%HRR)	30 mins/ 1 session	IC: ST * CF: WCST^00^
Chang (2014) [46]	Taiwan	NRS	5–10; M-85; DSM-IV-TR	27 (14/13)	8.19 ± 7.65 (8.78 ± 8.33)	I:4, HI:2, C:8 (I:3; C:10)	7 (6)	Aquatic exercise (control)	Moderate	90 mins/ 2 sessions/ 8 weeks	IC: GNG *
Chen (2022) [36]	China	RCT	6–10; M-82; DSM-5	64 (32/32)	8.37 ± 1.68 (7.89 ± 2.13)	I:6, HI:8, C:18 (I:5; HI:10; C:17)	none	Cycling (watching videos)	Moderate (60~80%HHR)	25 mins/ 3 sessions/ 12 weeks	IC: ST * CF: OEST * WM: N-B *
Chou (2017) [47]	Taiwan	NRS	8–12; M-77; psychiatric physician	49 (24/25)	10.71 ± 1.00 (10.30 ± 1.07)	I:12, HI:7, C:5 (I:8; HI:13; C:4)	10 (12)	Yoga (control)	Moderate (50~60%HHR)	40 mins/ 2 sessions/ 8 weeks	IC: DT *
Fritz (2022) [37]	USA	RCT	18–24; M-0; DMS-5	27 (11/16)	20.16 ± 1.46 (20.16 ± 1.46)	NR	NR	Yoga (control)	Moderate (RPE: 10.5–19)	180 mins/ 6 weeks	IC: FT^00^ CF: LSWMT^00^ WM: DCCST^00^
Hattabi (2019) [38]	Tunisia	RCT	9–12; M-87; DSM-IV	40 (20/20)	9.95 ± 1.31 (9.75 ± 1.33)	I:4, HI:6, C:10 (I:5; HI:4; C:11)	NR	Swimming (control)	Moderate (50–70%HRR)	90 mins/ 3 sessions/ 12 weeks	IC: ST *
Kouhbanani (2023) [40]	Iran	RCT	20–50; M-0; DSM-5	52 (25/27)	35.24 ± 11.49 (35.40 ± 11.01)	I:16; C:9 (I:18; C:9)	NR	Pilates (control)	Moderate	45 mins/ 3 sessions/ 24 weeks	CF: WCST *
Silva (2020) [43]	Brazil	RCT	11–14; M-70; DSM-IV	20 (10/10)	12.0 ± 2.0 (12.0 ± 1.0)	NR	NR	Swimming (control)	Moderate	45 mins/ 2 sessions/ 8 weeks	CF: TT *
Wu (2023) [45]	China	RCT	NR; M-66; psychiatric physician	83 (42/41)	9.81 ± 2.90 (9.92 ± 3.03)	I:16, HI:9, C:17 (I:14; HI:8; C:19)	NR	Cycling (watching videos)	Moderate (60%HHR)	20 mins/ 3 sessions/ 12 weeks	IC: ST * CF: OEST * WM: N-B *
Characteristics of included studies (Open skill).											
Kadri (2019) [39]	Tunisia	RCT	12–18; M-90; psychiatric physician	40 (20/20)	14.5 ± 3.5 (14.2 ± 3.0)	NR	NR	Taekwondo (control)	Moderate	50 mins/ 2 sessions/ 78 weeks	IC: SCWT *
Ludyga (2022) [41]	CH/DE	RCT	8–12; M-70; DSM-5	58 (29/29)	10.0 ± 1.2 (10.8 ± 1.2)	NR	29 (29)	Judo (control)	Moderate (RPE: 13.3 ± 0.7)	60 mins/ 2 sessions/ 12 weeks	WM: CDPT^00^
Pan (2016) [42]	Taiwan	RCT	6–12; M-100; DSM-IV	32 (16/16)	8.93 ± 1.49 (8.87 ± 1.56)	NR	9 (9)	Table tennis (control)	Moderate	70 mins/ 2 sessions/ 12 weeks	IC: SCWT *
Pan (2019) [48]	Taiwan	NRS	7–12; M-100; DSM-IV	60 (15/G1: 15, G2: 30TD)	9.08 ± 1.43 (8.90 ± 1.66)	NR	9 (9)	Table tennis (control)	Moderate	70 mins/ 2 sessions/ 12 weeks	IC: SCWT * CF: WCST *
Song (2022) [44]	China	RCT	6.58~8.58; M-100; DMS-IV	24 (8/G1: 8, G2: 8TD)	7.68 ± 0.56 (7.53 ± 0.79)	I:6, HI:0, C:2 (I:5; HI:1; C:2)	NR	Football (physical education class)	Moderate	60 mins/ 5 sessions/ 6 weeks	IC: SCWT * CF: TMT * WM: CFT *

CH = Switzerland, DE = Germany, RCT = randomized control trial, NRS = non-randomized studies, DSM-IV and-5 = Diagnostic and Statistical Manual of Mental Disorders, fourth edition and fifth edition, NR = no report, %HRR = percentage of heart rate reserve, I = ADHD-inattentive, HI = ADHD-hyperactive–impulsive, C = ADHD-combined, IC = inhibitory control, CF = cognitive flexibility, WM = working memory, ST = Stroop test, WCST = Wisconsin card sorting test, SCWT = Stroop color–word test, GNG = go no-go, OEST = odd even size test, N-B = N-back test, DT = determination test, RPE = Rated Perceived Exertion, FT = Flanker task, LSWMT = list sorting working memory test, DCCST = dimensional change card sort test, TT = trails test, CDPT = change detection paradigm test, TMT = trail making test, CFT= complex figure test. * = significant statistical improvement, ^00^: no statistically significant change.

**Table 2 behavsci-14-00499-t002:** ROBINS-I (risk of bias judgments in non-randomized studies of interventions).

Author (Year)	Confounding	Selection of Participants	Classification of Interventions	Deviations from Intended Interventions	Missing Data	Measurement of Outcomes	Selection of Reported Results	Overall
Chang (2014) [46]	Moderate	Low	Low	Low	Moderate	Moderate	Moderate	Moderate
Chou (2017) [47]	Moderate	Low	Low	Low	Moderate	Low	Low	Moderate
Pan (2019) [48]	Moderate	Low	Low	Low	Low	Moderate	Low	Moderate

**Table 3 behavsci-14-00499-t003:** Moderating analysis of different intervention features on each sub-function.

Categories	Subgroup	Outcome Variables	Q_b_	*p*	Combined Effects Test			Heterogeneity Test		
					SMD	95% CI	*p*	Q_w_	*p*	I^2^ (%)
Type of skills	Closed skill Open skill	Inhibitory control	6.263	0.012 *	−1.020	−1.464, −0.578	0.000	20.573	0.002	70.8
					−2.042	−2.710, −1.376	0.000	6.366	0.095	52.8
		Cognitive flexibility	0.256	0.613	−1.384	−2.324, −0.443	0.004	58.906	0.000	91.5
					−1.039	−1.983, −0.095	0.031	2.042	0.153	51.0
		Working memory	0.440	0.507	−0.865	−1.231, −0.498	0.000	2.614	0.271	23.5
					−0.521	−1.467, 0.425	0.280	2.735	0.098	63.4
Frequency	≤2 sessions/ week >2 sessions/ week	Inhibitory control	0.003	0.959	−1.344	−2.002, −0.685	0.000	31.676	0.000	81.0
					−1.367	−1.996, −0.738	0.000	10.881	0.012	72.4
		Cognitive flexibility	0.117	0.732	−1.171	−2.322, −0.021	0.046	22.764	0.000	86.8
					−1.442	−2.479, −0.404	0.006	31.360	0.000	90.4
		Working memory	7.082	0.008 **	−0.243	−0.673, 0.186	0.267	0.534	0.465	0.0
					−0.975	−1.301, −0.650	0.000	1.510	0.470	0.0
Period	<12 weeks ≥12 weeks	Inhibitory control	8.637	0.003 **	−0.759	−1.118, −0.400	0.000	4.768	0.311	16.1
					−1.833	−2.452, −1.213	0.000	23.845	0.000	79.0
		Cognitive flexibility	0.094	0.759	−1.475	−2.835, −0.115	0.034	25.259	0.000	88.1
					−1.212	−2.202, −0.223	0.016	33.554	0.000	91.0
		Working memory	0.005	0.944	−0.712	−1.338, −0.086	0.026	0.923	0.337	0.0
					−0.681	−1.260, −0.103	0.021	8.201	0.017	75.6
Age	<18	Executive functions	7.031	0.008 **	−1.305	−1.676, −0.934	0.000	114.768	0.000	83.4
	≥18				−0.613	−0.965, −0.260	0.001	1.381	0.710	0.0

Note: Q_b_: between-group effect, Q_w_: homogeneity test within each group. * Indicates *p* < 0.05, significant difference; ** indicates *p* < 0.01, highly significant difference.

## Data Availability

This is an evidence synthesis study; all data are available from the primary research studies or can be circulated from the corresponding author.

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
