# Peer review of "Effects of Practicing Closed- vs. Open-Skill Exercises on Executive Functions in Individuals with Attention Deficit Hyperactivity Disorder (ADHD)—A Meta-Analysis and Systematic Review"

_behavsci, 2024, doi:10.3390/bs14060499_

Round 1

Reviewer 1 Report

Comments and Suggestions for Authors

The review is a well-organized literature study, which results in a very practical outcome: the suggestion on the type and intensity of physical exercises, improving executive functions in ADHD patients. I believe that this review will be very useful for clinicians, helping them to balance physical and pharmacological interventions for ADHD patients.

A minor point to improve would be a larger inclusion of other populations, such as European or African - but I understand what there is a lack of original research papers.

Also, it would be helpful to exemplify the two types of physical activities studied - i.e., the OSE and CSE - or to put them in a special table.

Comments on the Quality of English Language

The language might be improved by shortening and clarifying sentences.

Reviewer 2 Report

Comments and Suggestions for Authors

This is a compelling meta-analysis paper about improvement in executive functions in ADHD patients with some ways of intervention classified as open exercises or close exercises. It is a well written and good quality paper, although I recommend authors review some aspects:

1. Tables and figures in general require more detail and some formatting to be understood, what measure is in the y-axis, in figs. 3 and 4, and what means the dotted line, for example.  Or what is 1, 2 and 3 in those same figures in the x-axis.

2. The analyses performed described in figs. 2 and 3 and table 2 are not quite well described in the analysis section. I suggest it should be described thinking in the possibility of someone else being able to duplicate the analyses.

3. Table 1.2 head is wrong, it should be corrected.

4. The explanation in Table 4 seems quite arbitrary, I suggest authors explain a bit more for the understanding of the readers.

Reviewer 3 Report

Comments and Suggestions for Authors

Overall, I found the manuscript well written and informative. There are some areas that could be improved, in the introduction section.T

The introduction provides a comprehensive overview of ADHD, including its prevalence, impact on cognitive function and current treatment modalities. It moves well into a discussion of the potential benefits of physical activity for children with ADHD and lays the groundwork for systematic review and meta-analysis. However, several areas could be improved to increase the clarity and direction of the topic:

1. Ensure smooth transitions between sections. For example, the transition from discussing the prevalence of ADHD to the impact on cognitive function could be smoother.

2. If necessary, introduce subsections to guide the reader through different topics (e.g. prevalence, impact on cognitive function, treatment modalities, role of exercise).

3. The introduction should be more concise in some areas. For example, the detailed description of the side effects of drug treatments could be summarised, as the focus of the paper is on exercise-related interventions.

4. Focus more on why it is crucial to understand the effects of different types of exercise (OSE vs. CSE) on executive functions in ADHD. This defines a clearer purpose for the systematic review.
